# Theoretical Prediction Method for Erosion Damage of Horizontal Pipe by Suspended Particles in Liquid–Solid Flows

**DOI:** 10.3390/ma14154099

**Published:** 2021-07-23

**Authors:** Guoqiang Liu, Wenzhe Zhang, Liang Zhang, Jiarui Cheng

**Affiliations:** 1School of Mechanical Engineering, Baoji University of Arts and Sciences, Baoji 721016, China; liuguoqiang0504@163.com; 2Shaanxi Key Laboratory of Advanced Manufacturing and Evaluation of Robot Key Components, Baoji 721016, China; 3Well Testing Company, CNPC Xibu Drilling Engineering Company Limited, Karamay 834000, China; xjsyzhwzh@cnpc.com.cn; 4PetroChina Coalbed Methane Company Limited, Beijing 100028, China; xzhangliang@petrochina.com.cn; 5School of Mechanical Engineering, Xi’an Shiyou University, Xi’an 710065, China

**Keywords:** liquid–solid suspension flow, horizontal pipe erosion, erosion prediction model

## Abstract

In order to study the erosion of a pipe wall via a liquid–solid suspension flow, a two-phase flow model combined with an erosion forecasting model for multiparticle impact on horizontal pipe wall surfaces was established in this work on the basis of low-cycle fatigue theory. In the model establishment process, the effects of particle motion and material damage were considered, and a simplified method for predicting horizontal wall erosion was obtained. The calculated results showed that the particles impact the wall at a small angle of most liquid flow velocities, causing cutting erosion damage of the wall. The settling velocity and fluctuating velocity of the particles together determine the radial velocity of the particles, which affects the impact angle of the particles. The cutting erosion caused by the small-angle impact of the particles in the pipe is more likely to cause rapid loss of the wall material. Therefore, the pipe wall is usually evenly thinned.

## 1. Introduction

The constant impact of particles in a multiphase flow on walls may cause the deformation or mass loss of wall material. This phenomenon is commonly referred to as particle erosion. Severe particle erosion causes serious problems, such as wall thickness reduction, structural fracture, and equipment failure, as well as other issues that affect production safety. Particle erosion may involve one step (cutting failure) or multiple steps (fatigue failure) [1,2,3]. Multistep particle erosion comprises extrusion, stretching, fracture, and shedding processes. It can only be analyzed by simplifying the fatigue abrasion behavior of plastic metal under repeated impact load given its complexity [4,5].

Researchers have proposed more than 30 scouring models for the prediction of particle erosion in plastic and brittle materials since Finnie [6] first systematically proposed the theory of plastic material erosion in 1958. Tilly [7] divided the scouring process into two steps on the basis of experimental measurements. First, the impact of particles on the material surface causes the material to press toward the periphery. Then, the material is squeezed again and peeled off through subsequent particle impacts. Studies on multiparticle erosion can refer to fatigue failure research given that the mass loss of plastic materials is usually the result of multiple particle impacts.

The fatigue life of plastic materials is usually divided into short-lifetime and long-lifetime zones. Plastic strain plays a leading role in the long-lifetime zone. The material failure loading cycle in the long-lifetime zone is short and belongs to the high-load and low-cycle fatigue category. Elastic strain plays a leading role in the short-lifetime zone. The small initial contact area between particles and a wall results in remarkably high contact stress. Therefore, multiparticle erosion is generally classified into the low-cycle fatigue research category. Guo [8] performed a round-head impact test to measure the erosion rate of materials. Their experimental results showed that the minimum particle impact numbers that cause material loss at impact velocities of 2 and 15 m·s^−1^ are 1585- and 4-fold, respectively. This result shows that the material loss process approaches cutting failure when the particle impact velocity is high. Hutchings [9] used low-cycle fatigue theory to establish an erosion model on the basis of dynamic hardness and ductility, and they obtained the velocity index of circular particles perpendicular to the wall. Huang [10] also established a multiparticle erosion prediction model on the basis of low-cycle fatigue theory. This model has numerous considerations and heavy experimental workload because it contains multiple physical quantities and empirical parameters. Nsoesie [11] presented the analytical modeling of erosion behavior of Stellite alloys under solid-particle impact. Using their modified S-K model, for any of the Stellite alloys studied in this research, the erosion rate at the particle impact velocities of 84 and 98 m·s^−1^ can be predicted for different particle impinging angles. In addition, with this model, for any Stellite alloys that have a similar chemical composition to one of the alloys studied in this research, the erosion rate can be estimated. Carach [12] identified significant technological factors and investigated their impact on the machined surface quality of difficult-to-machine materials. Nunthavarawong [13] presented a new semi-empirical approach to estimate the wear coefficient of the material itself as a function of impact angles via an erosion test apparatus. The result showed that the effect of the different angles of impact on the impact energy can also represent the failure modes in the tool steel material as ductile failures.

In addition to the study of Huang, associated research in the past 20 years has shifted attention from the deduction of a theoretical model to the numerical calculation of erosion rate. The numerical calculation of erosion rate is often accompanied with the tracking of particles and liquid computation of the flow field. The multi-phase numerical calculation results are brought into the semi-empirical erosion formula to calculate the erosion rate of a special structural wall. The calculation of particle erosion generally includes various steps, including liquid computation of the flow field, computation of the inter-phase acting force (Euler–Lagrange Model) or volume fraction (Euler–Euler Model), computation of the second-phase motion, computation of the impact parameters between the second phase and wall, and computation of the material erosion rate [14]. Among them, liquid computation, computation of the inter-phase acting force, and computation of the second-phase motion provide essential parameters to calculate erosion rate. The material properties of walls and roughness have been considered in the latest commercial calculation software. According to these numerical calculation steps, the wall erosion rate of bending pipes, suddenly shrinking or suddenly expanding pipes, and complicated geometric structures has been predicted in many studies since 1990.

Currently, particle motion and impact wall parameters in liquid–solid flows are mainly gained through numerical calculation. First, the continuity equation and momentum equation of different phases are solved in discrete mode, thus obtaining the refined migration and impact wall parameters of particles. The Euler–Lagrange model has been widely applied in the calculation of contact between particles and a liquid, contact between particles, and contact between particles and a wall. However, the liquid numerical calculation of the multi-phase solution is a complicated process and some calculation steps have to depend on commercial software. The erosion rate can only be calculated by the independently compiled UDF codes. In practical applications, it is difficult to add extension codes in commercial software if plastic deformations of walls, electrochemical reactions of surfaces, and fatigue failures of materials are considered. Even it is feasible, it claims significantly heavy computing loads.

A relatively simple analytic solving method was applied in the calculation of particle flow in this study, through which the motion parameters and impact parameters of particles were gained. Finally, a simple algorithm for the erosion rate of pipe walls via a liquid–solid suspension flow was established. Combining with the flow test results, application conditions of the calculation model and the variation law of the calculated values were disclosed.

## 2. Calculation Methods

Liquid–solid flows can be divided into sediment flow and suspension flow according to the particle distribution on the tubular section. In a tubular liquid–solid flow system, the sedimentation of particles is mainly determined by the critical sedimentation velocity of particles. The main characteristics of a sediment flow and suspension flow are introduced as follows: (1) under the same flow velocity, the contact characteristics between particles and a wall in a sediment flow and suspension flow are different. Particles in a suspension flow mainly impact the wall intermittently, while particles in a sediment flow mainly slide continuously or roll on the wall. (2) All particles in the center of a suspension flow may impact the wall, but only few particles in a sediment flow are blocked by deposited bottom particles from direct contact with the wall. (3) The impact interval of particles in a suspension flow onto the wall is related to particle concentration and particle velocity. However, the contact time of particles in a sediment flow and the wall is sensitive to particle–particle distance and sliding velocity. The movement of particles in a suspension flow in Figure 1 was constructed. On this basis, the impact parameters of particles in a tubular flow onto the wall were deduced theoretically.

### 2.1. Calculation of Laminar and Turbulent Velocities of Liquid

The axial motion equation of the laminar flow of liquid was constructed by a cylindrical coordinate system:(1)$$\frac{\partial^{2}u_{c}}{\partial y^{2}}+\frac{\partial^{2}u_{c}}{\partial z^{2}}=\frac{\partial^{2}u_{c}}{\partial r^{2}}+\frac{1}{r}\frac{\partial u_{c}}{\partial r}+\frac{1}{r^{2}}\frac{\partial^{2}u_{c}}{\partial \theta^{2}}$$

As $\frac{dp}{dx}=\mu \left(\frac{\partial^{2}u_{c}}{\partial y^{2}}+\frac{\partial^{2}u_{c}}{\partial z^{2}}\right)=-\frac{\Delta p}{L}$, Equation (1) can be rewritten as:(2)$$\frac{d^{2}u_{c}}{dr^{2}}+\frac{1}{r}\frac{du_{c}}{dr}+\frac{\Delta p}{\mu L}=0$$

Based on integration, it gains $u_{c}=C_{1}\ln r-\frac{\Delta pr^{2}}{4\mu L}+C_{2}$. The laminar velocity of liquid reaches the maximum and ln*r* reaches the minimum at *r =* 0. To make the maximum of *u_c_* a fixed value, *C*_1_ = 0. *u_c_* = 0 at *r = d_l_/*2, so *C*_2_ = Δ*pd_l_*^2^/16 *μ·L_c_*. Therefore, the axial laminar velocity distribution of liquid in a pipe is:(3)$$u_{c}=\frac{\Delta p}{4\mu L}\left(\frac{d_{l}^{2}}{4}-r^{2}\right)$$

According to the Hagen–Poiseuille law, $\Delta p=\lambda \frac{L}{d_{l}}\frac{\rho_{c}Q_{c}^{2}}{2\pi^{2}r_{l}^{4}}=\frac{8\mu LQ_{c}^{2}}{\pi^{2}r_{l}^{6}u^{a}}$. The relation equation between velocity and flow rate can be expressed as:(4)$$u_{c}=\frac{2Q_{c}^{2}}{\pi^{2}r_{l}^{6}u^{a}}\left(\frac{d_{l}^{2}}{4}-r^{2}\right)$$

Cylindrical metal pipes mainly belong to hydraulic rough pipes after particle impact. Hence, it can be known that $\frac{u^{a}}{u^{*}}=2.5\ln\frac{r_{l}}{e}+4.75$ according to the integration of the Nikuradse experimental fitting relation $\frac{\bar{u}}{u^{*}}=2.5\ln\frac{r_{l}-r}{e}+8.5$. Based on the subtraction of these two equations, we obtain:(5)$$\frac{\bar{u}}{u^{*}}=\frac{u^{a}}{u^{*}}+3.75+2.5\ln(1-\frac{r}{r_{l}})$$
(6)$$u^{*}=\sqrt{\tau_{w}/\rho_{c}}=\tau_{w}=\sqrt{0.03{\left(u^{a}\right)}_{}^{7/4}{\left(\frac{\mu }{\rho R_{s}}\right)}^{1/4}}$$

According to Equation (5), the near-wall friction velocity of liquid is:

Therefore, Equation (4) can be rewritten as:(7)$$u_{c}=u^{a}+\left[3.75+2.5\ln(1-\frac{r}{r_{l}})\right]\cdot \sqrt{0.03{\left(u^{a}\right)}_{}^{7/4}{\left(\frac{\mu }{\rho_{c}r_{l}}\right)}^{1/4}}$$
where the average velocity is *u^a^* = *Q_c_*/π*r_l_*^2^.

### 2.2. Calculation of Axial and Radial Impact Velocities of Particles onto the Wall

Particles are influenced by the drag force of liquid, pressure gradient force, staff lift force, and virtual mass force in liquid, in addition to gravity, buoyancy, and mutual impact force. Whether the sliding behaviors of particles and a liquid are large is generally determined by a comparison of the Stokes number: $St=\left(r_{p}d_{p}^{2}u\right)/\left(18md_{l}\right)$. When *St* < 0.1, the relative difference rate of velocities between the liquid and particles is lower than 1%, and it is believed that the velocities of the liquid and particles are equal. When 0.1 < *St* < 1, the velocity difference between particles and the liquid is small. In this case, particle velocity can be viewed as consistent with liquid velocity when there are low requirements on calculation. When *St* > 1, there is a large sliding velocity between particles and the liquid. The liquid flow rate under different particle diameters with a following critical value of particles of *St* = 1 was calculated (Figure 2). When the particle diameter is smaller than 0.1 mm, most flow rate intervals of tubular liquid can follow the particle well. Under this circumstance, particle velocity is approximately equal to liquid velocity. When 0.1 mm < *dp* < 0.6 mm, particles with *uc* < 2 m·s^−1^ still have good following performance. In the flow rate interval of 2~20 m·s^−1^, the upper region of the curve reflects the significant difference between particle velocity and liquid velocity, while the lower region implies a good following performance of particles. When *dp* > 0.6 mm, there will be significant slippage between particles and the liquid.

Small particles can generally follow the liquid flow well, and the velocity of small particles is approximately equal to the liquid velocity. Hence, the liquid velocity can be used to replace the particle velocity to calculate the erosion rate. However, this is inapplicable to large particles or high-velocity particles, because of the sliding velocity between large particles or high-velocity particles and the liquid, particle buoyancy, gravity, and drag force, which can be expressed as:(8)$$F_{w}=\frac{\pi d_{p}^{3}}{6}\left(\rho_{p}-\rho_{c}\right)g$$
(9)$$F_{d}=\frac{\pi }{8}C_{d}\rho_{f}d_{p}^{2}\left(u_{c}-v_{p}\right)\left|u_{c}-v_{p}\right|$$

As the axial velocity component of liquid flow in a straight pipe is far higher than the radial velocity component, the axial component of the drag force to particles is also larger than the radial component. As a result, the impact process of particles onto the wall is mainly controlled by the drag force axially, but the radial impact process is mainly controlled by buoyancy and gravity together in laminar flow. The impact process of particles in turbulence is also related to the fluctuating velocity of liquid.

For particles with a radial distance of r to the center line, the radical impact displacement is *y = r_l_ − r − r_p_*. If the axial velocity of particles in contact with the wall is equal to the liquid velocity, the axial velocity of the liquid with a radial distance of *r* to the center line is:

Laminar: (10)$$u_{1x}=\frac{2Q_{c}^{2}}{\pi^{2}r_{l}^{6}u^{a}}\left(\frac{d_{l}^{2}}{4}-r^{2}\right)$$

Turbulent: (11)$$u_{1x}=u^{a}+\left[3.75+2.5\ln(1-\frac{r}{r_{l}})\right]\cdot \sqrt{0.03{\left(u^{a}\right)}_{}^{7/4}{\left(\frac{\mu }{\rho_{c}r_{l}}\right)}^{1/4}}$$

The average drag force when particles move from r to the wall surface is ${\bar{F}}_{d}=\frac{\pi }{8}C_{d}\rho_{c}d_{p}^{2}{\left(u_{1x}-u_{2x}\right)}^{2}$, where *u_2x_* is the axial liquid velocity in the boundary layer when particles contact with the wall. The drag force coefficient in the laminar region is *C_d_ =* 24*/Re_p_*, the turbulent resistance coefficient is *C_d_* = 0.44, and the Reynolds number of particles is *Re_p_* = *d_p_·ρ_c_·v*/*μ*. According to Newton’s second law, the axial accelerated velocity of particles is $a_{px}={\bar{F}}_{d}/m_{p}=\frac{\pi }{8}C_{d}\rho_{c}d_{p}^{2}{\left(u_{1x}-u_{2x}\right)}^{2}/m_{p}$. When the axial movement of particles is simplified into a uniformly retarded motion, if *St* < 1, the starting and final velocities of particles meet:(12)$$v_{2x}^{2}-v_{1x}^{2}=2a_{px}y=-\frac{\pi C_{d}\rho_{c}d_{p}^{2}{\left(u_{1x}-u_{2x}\right)}^{2}(r_{l}-r-r_{p})}{4m_{p}}$$
where the flow rate of liquid is equal to the particle velocity: *u*_1*x*_ = *v*_1*x*_. Therefore, the axial velocity when particles impact the wall is:(13)$$v_{2x}^{}=\sqrt{u_{1x}^{2}-\frac{\pi C_{d}\rho_{c}d_{p}^{2}{\left(u_{1x}-u_{2x}\right)}^{2}(r_{l}-r-r_{p})}{4m_{p}}}$$

The drag force coefficient varies with flow pattern. The fluid velocity at the initial position of particles is *u*_1*x*_ and the near-wall liquid velocity is *u*_2*x*_. Therefore, the axial velocity when particles impact the wall can be gained by bringing *u*_1*x*_ and *u*_2*x*_ into Equation (13).

The impact of particles onto the wall can be divided into two processes: (1) variable accelerated motion under changing stresses; (2) uniform motion after equilibrium resultant force. In the variable accelerated motion when particles move from the initial position to the wall surface, the radial displacement is *y* = *r*_l_ − *r* − *r_p_* and the radial accelerated velocity of particles is *a_py_* = *πd_p_*^3^(*ρ_p_* − *ρ_c_*)*g/*6 *m_p_*, which are similar to those in the axial calculation. Therefore, the radial velocity of particles in the accelerated motion region is:(14)$$v_{1y}=\sqrt{{\bar{v}}_{y}^{2}+\frac{\pi d_{p}^{3}g\left(\rho_{p}-\rho_{c}\right)\left(r_{l}-r-r_{p}\right)}{3m_{p}}\cos\theta \cos\beta }$$
where *v*_1*y*_ is the initial radial velocity of particles and *v*_1*y*_ = 0 m·s^−1^ for laminar flow. For turbulent flows, *v*_1*y*_ is the radial fluctuating velocity. In fact, the second term in Equation (14) is the additional radial velocity of buoyant weight of particles. As the drag force of liquid increases as a response to the accelerating sedimentation of particles in the radial uniform motion, fully deposited particles finally become stable. The final settling velocity under different Reynolds Numbers is:(15)Stokes region: vc=gdp2(ρp−ρc)18μ 10−4<Rep<2
(16)Allen region: vc=[4225×g2(ρp−ρc)2ρcμ]1/3⋅dp 2<Rep<500
(17)Newton region: vc=3.03dpg(ρp−ρc)ρc 500<Rep<2×105

Therefore, the radial particle velocity is a function of fluctuating velocity and settling velocity of particles in the uniform motion region:(18)$$v_{2y}=\sqrt{{\bar{v}}_{y}^{2}+v_{c}\cdot \cos\theta \cos\beta }$$

The fluctuating velocity in the above calculation formula of radial velocity is defaulted to 0 m·s^−1^ throughout the laminar flow process. Attention shall be paid to the calculation of whether particles reach the critical settling velocity only. When drag force is neglected and the critical settling velocity of particles is smaller than the final settling velocity in the process of turbulent flow, the actual radial impact velocity has to be calculated from Equation (14). If the critical settling velocity is higher than or equal to the final settling velocity, the actual radial velocity can be calculated from Equation (18). For round ceramic particles (*d_p_* = 0.6 mm) in this experiment, the Reynolds number is *Re_p_* = 600~6000 in the flow rate range of 1~10 m·s^−1^. Under this circumstance, particles deposit at the Newton region and the final settling velocity is 0.12 m·s^−1^. The final settling velocity in the Allen region where *Re_p_* < 500 is 0.064 m·s^−1^. In conclusion, the final settling velocity of 0.6 mm ceramic particles is very small, especially in the Allen region. As the maximum settling accelerated velocity is *a_py_ = πd_p_*^3^ (*ρp − ρc)g/*6 *m_p_* ≈ 4.7 m·s^−^^2^, particles should only move by 1.53 mm in the Allen region to reach the final settling velocity. This accelerating distance can be neglected compared with the pipe diameter. As a result, the radial accelerating movement of particles can be overlooked in this study and the radial impact velocity of particles can be expressed as Equation (18).

As the actual radial impact velocity of particles is the vector sum of the radial fluctuating velocity and settling velocity of particles, the particle velocity under different azimuth angles of the tubular section is significantly different. The impact velocity of particles on the lower semi-section of the pipe is the sum of fluctuating velocity and settling velocity, while the impact velocity on the upper semi-section is the difference between fluctuating velocity and settling velocity. In this case, particles that are expected to impact the upper pipe wall may deposit and then impact the lower pipe wall if the fluctuating velocity is smaller than the settling velocity. As a result, the erosion rate of the lower pipe wall is significantly higher than that of the upper pipe wall. When the tangential velocity gradient in the turbulence core region is neglected, that is, *∂u*/*∂y* = 0, the tangential stress on any surface in the turbulence is expressed by fluctuating velocity: $\tau =\rho_{c}{\bar{u}}_{x}{\bar{u}}_{y}$. According to the surface friction coefficient of liquid, it can also be expressed as:(19)$$C_{f}=2\tau /\rho_{c}{\left(u^{a}\right)}^{2}=\frac{1}{4}\lambda_{c}$$
where *λc* = 0.3164 *Re*^−0.25^ for a smooth cylindrical pipe with *Re* < 10^5^, and *λc* = 0.0008 + 0.055/*Re*^0.237^ for a smooth cylindrical pipe with 10^5^ < *Re* < 3 × 10^6^. Hence, the fluctuating velocity is ${\bar{u}}_{x}{\bar{u}}_{y}=0.125{\left(u^{a}\right)}^{2}\lambda_{c}$. According to Laufer [12], the radial fluctuating velocity and axial fluctuating velocity meet the relation of ${\bar{u}}_{x}=c{\bar{u}}_{y}$. It was measured in their experiment that the coefficient *c* is close to 1 approaching the center line, but it is close to 2 approaching the wall. Hence, *c* is determined as the mean of 1.5 for most particles in the radial position. In this way, the radial fluctuating velocity of liquid is ${\bar{u}}_{y}={\bar{v}}_{y}\approx u^{a}\sqrt{0.08\lambda_{c}}$ and it is substituted into Equation (18), obtaining the radial impact velocity of particles in a turbulent flow:(20)$$v_{y}=\sqrt{0.08\lambda_{c}{\left(u^{a}\right)}^{2}+v_{c}\cdot \cos\theta \cos\beta }$$

For liquid–solid flows in an inclined cylindrical pipe with a dip angle of *β*, the axial and radial impact velocities of particles when the liquid is Newtonian fluid are:

Laminar:(21)$$v_{x}^{}=\sqrt{{\left[\frac{2Q_{c}^{2}}{\pi^{2}r_{l}^{6}u^{a}}\left(\frac{d_{l}^{2}}{4}-r^{2}\right)\right]}^{2}-\frac{\pi C_{d}\rho_{c}d_{p}^{2}{\left(u_{1x}-u_{2x}\right)}^{2}(r_{l}-r-r_{p})}{4m_{p}}}+v_{c}\cdot \cos\theta \sin\beta$$
(22)$$v_{y}=v_{c}\cdot \cos\theta \cos\beta =\frac{4gd_{p}^{}\left(\rho_{p}-\rho_{c}\right)}{3\rho_{c}C_{d}}\cdot \cos\theta \cos\beta$$

Turbulent:(23)$$v_{x}^{}=\sqrt{\begin{array}{l} {\left\{u^{a}+\left[3.75+2.5\ln(1-\frac{r}{r_{l}})\right]\cdot \sqrt{0.03{\left(u^{a}\right)}_{}^{7/4}{\left(\frac{\mu }{\rho_{c}r_{l}}\right)}^{1/4}}\right\}}^{2}\\ -\frac{\pi C_{d}\rho_{c}d_{p}^{2}{\left(u_{1x}-u_{2x}\right)}^{2}(r_{l}-r-r_{p})}{4m_{p}} \end{array}}+v_{c}\cdot \cos\theta \sin\beta$$
(24)$$v_{y}=\sqrt{0.08\lambda_{c}{\left(u^{a}\right)}^{2}+v_{c}\cdot \cos\theta \cos\beta }$$

### 2.3. Calculation of the Pipe Wall Erosion under Continuous Particle Impact

As reported in our previous study on particle impact erosion [15,16], the erosion rate under tangential nonslipping particle impact can be expressed as follows:(25)$$\begin{array}{l} C_{E}=0\\ C_{E}=\frac{1}{3}(Lr_{p}^{1/2}{\left(2hN_{c}^{b}\right)}_{}^{3/2}+r_{p}{\left(2hN_{c}^{b}\right)}_{}^{2}) \end{array}\begin{array}{l} N_{c}<N\\ N_{c}\geq N \end{array}$$
where $N_{c}=\frac{3r^{2}u^{a}Q_{V}}{4r_{p}^{3}}$, $h=\frac{-B+\sqrt{B^{2}-2AC_{1}}}{2A}$, $L=\frac{-B+\sqrt{B^{2}-4AC_{2}}}{2A}$, $A=\frac{1}{2}\sigma_{y}\pi r_{p}^{}$, $B=0.17\times \frac{\pi^{3}\sigma_{y}^{3}r_{p}^{2}}{{\left(E^{*}\right)}^{2}}$, $C_{1}=-m_{p}v_{y0}^{2}$, and $C_{2}=-m_{p}v_{x0}^{2}$.

The erosion rate of pipe flow under tangential slipping particle impact can be expressed as follows:(26)$$\begin{array}{ll} C_{E}=0 & N_{c}<N\\ C_{E}=\frac{1}{3}Lr_{p}^{1/2}{\left(2hN_{c}^{b}\right)}^{3/2} & N_{c}\geq N \end{array}$$
where $N_{c}=\frac{3r^{2}u^{a}Q_{V}}{4r_{p}^{3}}$, $h=\frac{-B+\sqrt{B^{2}-2AC}}{2A}$, $F_{y}=\sigma_{y}\pi r_{p}h$, $F_{x}=\frac{v_{x0}\left(1-\lambda_{x}\right)}{2v_{y0}\left(1-\lambda_{y}\right)\sigma_{y}\pi r_{p}^{}h_{}^{2}}$, $A=\frac{1}{2}\sigma_{y}\pi r_{p}^{}$, $L=2\sqrt{r_{p}h}/{\left(1-\frac{F_{x}}{\mu F_{y}}\right)}^{1/3}$, $B=0.17\times \frac{\pi^{3}\sigma_{y}^{3}r_{p}^{2}}{{\left(E^{*}\right)}^{2}}$, and $C=-m_{p}v_{y0}^{2}$.

By combining the particle impact velocity model (Equations (21)–(24)) and the multi-particle impact erosion model (Equations (25) and (26)), the pipe erosion model, as shown in Appendix A, is obtained.

## 3. Results

### 3.1. Calculated Results of Impact Parameters

The axial impact velocity and radial impact velocity of particles onto the wall of a smooth cylindrical pipe (inner diameter = 40 mm) under different flow rates were calculated. The axial impact velocity (*v_x_*) and the radial impact velocity (${\bar{u}}_{y}$) were calculated according to the mean velocity (*u^a^*) and different phase angles corresponding to different radial settling velocities. Based on the calculated results, the radial fluctuating velocity is about 2 orders of magnitudes smaller than the axial fluctuating velocity under the same flow rate of turbulent flow. The maximum settling velocity under different phase angles is 1 m·s^−1^, which is about equal to the axial velocity under the flow rate of 8 m^3^·h^−1^. The settling velocity negatively correlates with the phase angle and it decreases to 0 m·s^−1^ at 90°, and further decreases to a negative value after 90°, which indicates the vertical downward direction (Figure 3). Therefore, it can be concluded that under low flow rate, the axial impact velocity of particles at small phase angles is close to the radial fluctuating velocity, while the radial fluctuating velocity is controlled by the settling velocity. At large phase angles, the axial impact velocity, radial fluctuating velocity, and settling velocity are close and the impact angle is large. Under high flow rates, the axial impact velocity is significantly higher than the radial fluctuating velocity and settling velocity, thus making particles impact the wall at an extremely small angle.

The relation curve between settling velocity and phase angle under different pipe diameters is shown in Table 1. The vertical downward direction reflects the position of *θ* = 0°. The positive and negative values imply the consistency between velocity direction and gravity direction. With the increase in pipe diameter, the settling velocity at *θ* = 0° increases gradually. Influences of changes in pipe diameter decrease gradually approaching the phase angle of 90°, indicating the consistent changes of pipe diameter and settling velocity. The vertical settling velocity is mostly influenced by changes in pipe diameter.

### 3.2. Pipe Erosion Rate

The flow of a 0.6 mm ceramic particle in water is analyzed in this section. The impact velocity and impact frequency of this 0.6 mm ceramic particle were calculated according to Section 2.1. The variation trend of erosion rate with phase angle and flow rate is shown in Figure 4. In the calculation function of the erosion rate of particles, there is a nonlinear relationship between erosion rate and number of particles, which determines the variation trend of erosion rate. The growth rate of erosion rate under a low flow rate is higher than that under a high flow rate. Specifically, the erosion rate when the flow rate is lower than 15 m·s^−1^ is basically smaller than 0.1 mm·h^−1^, but the maximum erosion rate reaches 0.61 mm·h^−1^ at 30 m·s^−1^. Due to different settling velocities at different phase angles and the maximum fluctuating velocity close to the settling velocity, the maximum erosion rate in suspension media reaches the maximum at the pipe bottom and decreases gradually approaching the pipe top. It can be seen from the results that the erosion rate increases by about 43.3% when the phase angle increases from 0° to 60°. This reveals that the erosion rates at different phase angles of the wall in a suspension flow are significantly different. Key attention should be paid to material loss at the bottom when predicting the wall thinning.

To better reflect the variation trend of erosion rate with phase angle and flow rate, flow rates in a 40 mm pipe were set to 0.02, 0.2, 2, and 20 m·s^−1^, and the phase angle range was set 0°~82°. Erosion rates of the super 13Cr stainless steel wall under these conditions were calculated (Table 2). In this study, the number of times the erosion rate was smaller than that of the flow rate was found. For example, the erosion rate only increased about 4.6 times when the flow rate increased by 10 times from 2 to 20 m·s^−1^. The number of times the erosion rate in other flow rate ranges was consistent was found. In other words, the growth of erosion rate was about 50% of that of the flow rate, and the growth ratio between erosion rate and flow rate was a fixed value.

According to impact angles at different phase angles, the radial impact velocity was faster than the axial impact velocity at a low flow rate (0.2 m·s^−1^), so the impact angles at most phase angles were larger than 45° and deep impact pits formed. When the flow rate increased to 2 m·s^−1^, the radial impact velocity was about 50% of that of the axial impact velocity, and the corresponding impact angle was smaller than 30°, which is easy to produce cutting erosion. When the flow rate increased to 20 m·s^−1^, the impact angle was smaller than 5°. In other words, all particles slid along the wall at a small angle. In most pipe flows, the flow rate was mainly 1 m·s^−1^ and particles mainly impacted the pipe wall at a small angle. Therefore, particles in tubular flows mainly impacted the pipe wall through micro-cutting failures.

## 4. Discussion

As shown in Figure 5, the experimental loop mainly comprised a perforating test section, a screw pump (the flow range is 1~14 m^3^/h), an electric heating agitator, a temperature and a pressure sensor, a magnetic flowmeter (8712HR, Rosemount. Co., Shakopee, MN, USA), three gate values, a control cabinet, a computer, and two flow pipes. When the perforating fluid containing sand was mixed well, the stirring and electric heating were turned on until the temperature reached a desired value, and the pump was then opened. Gate valve 3 in the test pipe was opened when the flow reached stability, and related data, including particle motion image, flow pressure, and temperature, were recorded.

In order to facilitate photographic documentation, the test device was made of organic glass. It also consisted of two switching pipe sections (Figure 6a). There were five samples with a diameter of 4 mm in each cross-section (Figure 6b), and they were arranged at an angle of 45°, which represented the different phase angles. The samples were made of 13Cr stainless steel, and its composition and mechanical properties are shown in Table 3 and Table 4. The exposed surface was sealed with epoxy resin and ground using SiC emery paper grade 1200 prior to installation. The sample surface profiles were verified via H1200WIDE confocal scanning laser microscopy (Lasertec. Co., Ltd., Yokohama, Japan).

Experimental results and calculated results under different phase angles at a fixed flow rate of 0.6 m·s^−1^ are listed in Table 5. According to the experimental results, the material loss at the tube bottom (phase angle = 0°) was about 8.6 times that at the tube top (phase angle = 180°). However, the difference in material loss between the tube bottom and tube top was even larger, indicating the evident deviation of calculated results. It can also be found from the relative error between calculated results and experimental results at different positions that the calculated results of erosion rates at the tube bottom and tube top deviated from the experimental results. The deviation at the tube bottom was positive, indicating that the calculated results were higher than the experimental results. The deviation at the tube top was negative, indicating that the experimental results were higher than the calculated results. According to the critical settling velocity of particles, which was calculated according to Figure 3, most particles deposited when the flow rate was 0.6 m·s^−1^. At this moment, a sedimentation layer formed at the bottom, and particles mainly maintained continuous friction contact with the wall rather than instantaneous impact contact, thus resulting in the large calculation error. As few particles impacted the tube top, the calculation error of the number of particles amplified. In the experiment, some small particles or broken particles impacted the upper wall due to the uneven particle diameter. These particles were not considered in the calculation, which further caused a negative deviation. The calculation error was relatively small (14.74%) at a 90° phase angle (horizontal position). This implies that the proposed calculation method had a high accuracy to the wall loss close to the horizontal surface under sediment flows.

Particles were distributed on the tubular section more uniformly by increasing the flow rate, thus realizing the suspension flow conditions. The corresponding calculated results and experimental results are listed in Table 6. With the increase in flow rate, the calculation error at the tube bottom decreased gradually. The calculation errors at 1.8 and 1.2 m·s^−1^ changed greatly, indicating the high prediction accuracy at the tube bottom under a turbulent flow of particles. In other words, the proposed prediction method was more applicable to suspension flow.

## 5. Conclusions

The two-phase flow model combined with the erosion forecasting model for multi-particle impact on pipe wall surfaces was established in this work on the basis of low-cycle fatigue theory. By comparing the test and the calculation results, the following conclusions were obtained.

The axial velocity of the particles was greater than the radial velocity at most liquid flow velocities in the horizontal pipe, which caused the particles to impact the wall at a small angle. As a result, the impact angle of the particles and the wall decreased with the flow velocity.

(1)When the settling velocity of the particles is greater than the fluctuating velocity of the particles, there is a significant difference in the erosion rate along the circumferential direction of the pipe wall; if not, there are no significant differences.(2)The experimental results showed that the existing model has a large error in predicting the erosion of the top and bottom walls of the pipe. If you can add material-related empirical coefficients to the model, or evenly suspend the particles, this error can be reduced.

## Figures and Tables

**Figure 1 materials-14-04099-f001:**
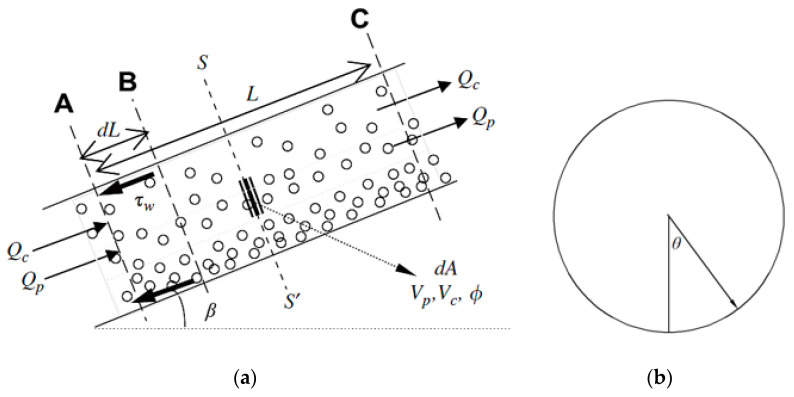
Movement of particles in liquid–solid suspension flow: (**a**) front view; (**b**) profile map.

**Figure 2 materials-14-04099-f002:**
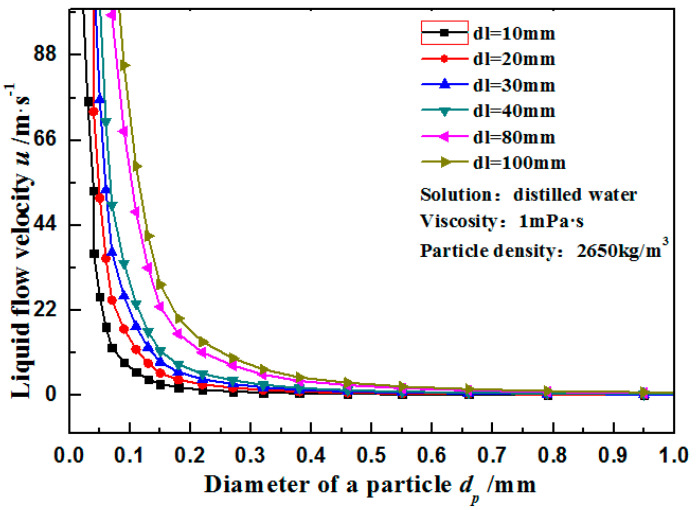
Distributions of pipe diameter and particle diameter when *St* = 1.

**Figure 3 materials-14-04099-f003:**
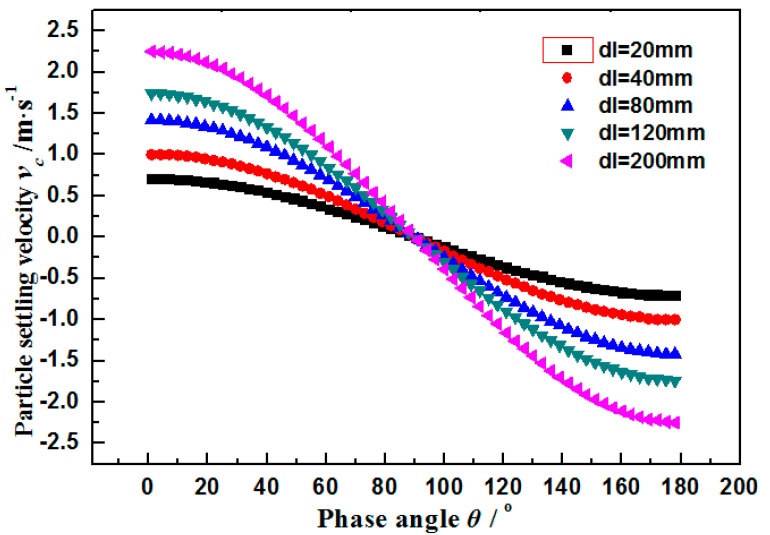
Relationship between settling velocity and phase angle under different pipe diameters.

**Figure 4 materials-14-04099-f004:**
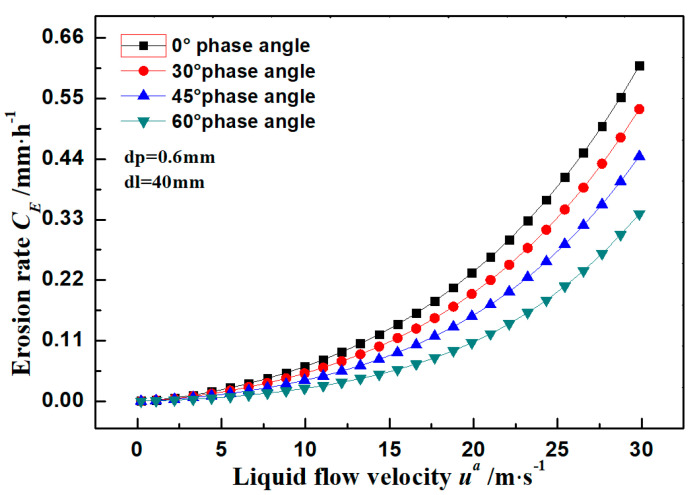
Erosion rates of pipe wall at different phase angles.

**Figure 5 materials-14-04099-f005:**
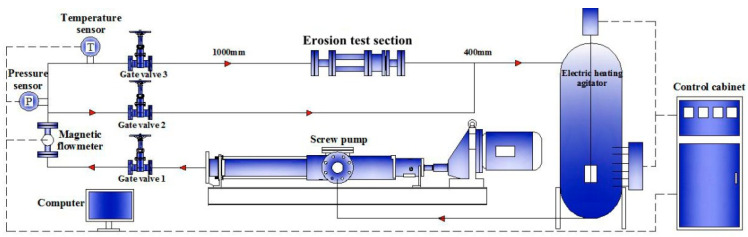
Schematic diagram and actual picture of experimental setup.

**Figure 6 materials-14-04099-f006:**
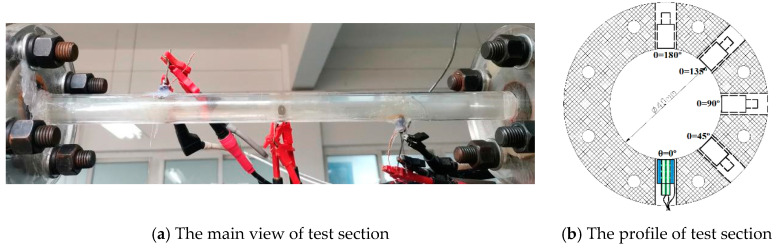
Schematic diagram of pipe flow erosion test section (*θ* = 0°).

**Table 1 materials-14-04099-t001:** Calculated results of particle velocity components in a 40 mm cylindrical pipe under different flow rates (*d_p_* = 0.6 mm, *ρ*_*p*_ = 1850 kg·m^−3^).

*Q_c_* (m^3^·h^−1^)	*u^a^* (m·s^−1^)	*v_x_* (m·s^−1^)	$\bar{u}$_*y*_ (m·s^−1^)	*θ* (°)	*v_c_* (m·s^−1^)
1	0.22	0.15	3.15 × 10^−5^	1	1.00
5	1.11	0.81	7.84 × 10^−4^	4	1.00
10	2.21	1.66	3.13 × 10^−3^	7	1.00
15	3.32	2.53	7.05 × 10^−3^	10	0.99
20	4.42	3.41	0.01	13	0.98
25	5.53	4.30	0.02	16	0.97
30	6.63	5.20	0.03	19	0.95
35	7.74	6.09	0.04	22	0.93
40	8.85	6.99	0.05	25	0.91
45	9.95	7.90	0.06	28	0.89
50	11.06	8.81	0.08	31	0.86
55	12.16	9.72	0.09	34	0.83
60	13.27	10.63	0.11	37	0.80
65	14.38	11.54	0.13	40	0.77
70	15.48	12.46	0.15	43	0.73
75	16.59	13.38	0.18	46	0.70
80	17.69	14.30	0.20	49	0.66
85	18.80	15.22	0.23	52	0.62
90	19.90	16.14	0.25	55	0.58
95	21.01	17.06	0.28	58	0.53
100	22.12	17.99	0.31	61	0.49
105	23.22	18.91	0.35	64	0.44
110	24.33	19.84	0.38	67	0.39
115	25.43	20.77	0.41	70	0.34
120	26.54	21.70	0.45	73	0.29
125	27.65	22.63	0.49	76	0.24
130	28.75	23.56	0.53	79	0.19
135	29.86	24.49	0.57	82	0.14

**Table 2 materials-14-04099-t002:** Erosion rates under different phase angles and different flow rates.

Phase Angles (*θ*/°)	Erosion Rates (C_E_/mm·h^−1^)				Particle Impact Angles (α/°)		
	0.02/m·s^−1^	0.2/m·s^−1^	2/m·s^−1^	20/m·s^−1^	0.2/m·s^−1^	2/m·s^−1^	20/m·s^−1^
1	1.02 × 10^−3^	5.75 × 10^−3^	3.23 × 10^−2^	1.82 × 10^−1^	78.74	26.67	2.88
4	1.32 × 10^−3^	7.42 × 10^−3^	4.17 × 10^−2^	2.34 × 10^−1^	78.72	26.62	2.87
7	1.06 × 10^−3^	5.94 × 10^−3^	3.34 × 10^−2^	1.88 × 10^−1^	78.66	26.50	2.85
10	9.08 × 10^−4^	5.11 × 10^−3^	2.87 × 10^−2^	1.61 × 10^−1^	78.57	26.32	2.83
13	8.04 × 10^−4^	4.52 × 10^−3^	2.54 × 10^−2^	1.43 × 10^−1^	78.45	26.08	2.80
16	7.22 × 10^−4^	4.06 × 10^−3^	2.28 × 10^−2^	1.28 × 10^−1^	78.30	25.78	2.76
19	6.54 × 10^−4^	3.68 × 10^−3^	2.07 × 10^−2^	1.16 × 10^−1^	78.11	25.41	2.72
22	5.96 × 10^−4^	3.35 × 10^−3^	1.89 × 10^−2^	1.06 × 10^−1^	77.88	24.98	2.67
25	5.45 × 10^−4^	3.06 × 10^−3^	1.72 × 10^−2^	9.69 × 10^−2^	77.61	24.48	2.61
28	4.99 × 10^−4^	2.81 × 10^−3^	1.58 × 10^−2^	8.87 × 10^−2^	77.30	23.92	2.54
31	4.57 × 10^−4^	2.57 × 10^−3^	1.45 × 10^−2^	8.13 × 10^−2^	76.93	23.30	2.47
34	4.19 × 10^−4^	2.36 × 10^−3^	1.33 × 10^−2^	7.46 × 10^−2^	76.50	22.61	2.39
37	3.85 × 10^−4^	2.16 × 10^−3^	1.22 × 10^−2^	6.84 × 10^−2^	76.01	21.86	2.30
40	3.53 × 10^−4^	1.98 × 10^−3^	1.12 × 10^−2^	6.27 × 10^−2^	75.43	21.05	2.20
43	3.23 × 10^−4^	1.82 × 10^−3^	1.02 × 10^−2^	5.74 × 10^−2^	74.77	20.17	2.10
46	2.95 × 10^−4^	1.66 × 10^−3^	9.33 × 10^−3^	5.25 × 10^−2^	74.01	19.24	2.00
49	2.69 × 10^−4^	1.51 × 10^−3^	8.51 × 10^−3^	4.79 × 10^−2^	73.12	18.24	1.89
52	2.45 × 10^−4^	1.38 × 10^−3^	7.74 × 10^−3^	4.35 × 10^−2^	72.08	17.19	1.77
55	2.22 × 10^−4^	1.25 × 10^−3^	7.01 × 10^−3^	3.94 × 10^−2^	70.86	16.07	1.65
58	2.00 × 10^−4^	1.12 × 10^−3^	6.32 × 10^−3^	3.55 × 10^−2^	69.41	14.91	1.53
61	1.79 × 10^−4^	1.00 × 10^−3^	5.65 × 10^−3^	3.18 × 10^−2^	67.68	13.69	1.40
64	1.58 × 10^−4^	8.90 × 10^−4^	5.01 × 10^−3^	2.82 × 10^−2^	65.58	12.42	1.26
67	1.38 × 10^−4^	7.79 × 10^−4^	4.38 × 10^−3^	2.46 × 10^−2^	63.00	11.11	1.12
70	1.19 × 10^−4^	6.68 × 10^−4^	3.76 × 10^−3^	2.11 × 10^−2^	59.80	9.75	0.98
73	9.91 × 10^−5^	5.58 × 10^−4^	3.14 × 10^−3^	1.76 × 10^−2^	55.75	8.36	0.84
76	7.90 × 10^−5^	4.44 × 10^−4^	2.50 × 10^−3^	1.41 × 10^−2^	50.55	6.93	0.70
79	5.78 × 10^−5^	3.25 × 10^−4^	1.83 × 10^−3^	1.03 × 10^−2^	43.79	5.48	0.55
82	3.42 × 10^−5^	1.92 × 10^−4^	1.08 × 10^−3^	6.08 × 10^−3^	34.96	4.00	0.40

**Table 3 materials-14-04099-t003:** Chemical composition of 13Cr steel (wt%).

Materials	C	Si	Mn	P	S	Cr	Mo	Ni	Cu
13Cr	0.029	0.22	0.45	0.015	0.001	13.3	1.92	4.85	1.59

**Table 4 materials-14-04099-t004:** Mechanical properties of 13Cr steel.

Materials	Tensile Strength (MPa)	Yield Strength (MPa)	Elongation (%)	Hardness (HV)
13Cr	970	855	20	323.4

**Table 5 materials-14-04099-t005:** Comparison between experimental results and calculated results at different phase angles (flow velocity = 0.6 m·s^−1^).

Phase Angle (*θ*/deg)	Experimental Value (C_E_/mm·h^−1^)	Calculated Value (C_E_/mm·h^−1^)	Relative Error (%)
0	1.20 × 10^−2^	1.49 × 10^−2^	24.17
45	8.30 × 10^−3^	9.62 × 10^−3^	15.90
90	5.70 × 10^−3^	6.54 × 10^−3^	14.74
135	2.10 × 10^−3^	1.84 × 10^−3^	−12.38
180	1.40 × 10^−3^	8.01 × 10^−4^	−42.79

**Table 6 materials-14-04099-t006:** Comparison between experimental results and calculated results at tube bottom under different flow rates (phase angle = 0°).

Flow Velocity (m·s^−1^)	Experimental Value (C_E_/mm·h^−1^)	Calculated Value (C_E_/mm·h^−1^)	Relative Error (%)
0.6	1.20 × 10^−2^	1.49 × 10^−2^	24.17
1.2	1.70 × 10^−2^	1.95 × 10^−2^	21.76
1.8	2.30 × 10^−2^	2.63 × 10^−2^	14.35
2.4	3.16 × 10^−2^	3.51 × 10^−2^	11.08

## Data Availability

Not applicable.

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
