# Peer review of "Theoretical Prediction Method for Erosion Damage of Horizontal Pipe by Suspended Particles in Liquid–Solid Flows"

_materials, 2021, doi:10.3390/ma14154099_

Round 1

Reviewer 1 Report

Manuscript deals with theoretical prediction method for erosion damage of Horizontal Pipe by Suspended Particles in Liquid-Solid Flows which is important both from the practical point of view and theoretical point of view of material wear with. Generally, manuscript fits the journal scope and it can be considered for publication after consideration of following suggestions:

  • In the Introduction literature review, each citation should be done individually for a single reference. Clubbing of more than one referred articles​ ​by one single statement for citation as it is done in several cases should be avoided otherwise it would be inferred that citations are done only for the formality without having focused and precise relevance. (see paragraph: Particle erosion may involve one step (cutting failure) or multiple steps (fatigue 29 failure) [1-3]. Multistep particle erosion comprises extrusion, stretching, fracture, and 30 shedding processes. It could only be analyzed by simplifying the fatigue abrasion behavior of plastic metal under repeated impact load given its complexity [4,5].
  • The introduction is suggested to update with recent published works dealing with wear of materials
  • Write up for the research gap should be in a separate single paragraph to create the appropriate prelude for the motivation of the work.
  • Authors presented the results without relevant discussion. Without discussion it is not possible to recognise what is original work of the authors and what is already generally known to the scientific community. Discuss what your results may mean for researchers in the same field as you, researchers in other fields, and the general public. How could your findings be applied or how your study improves the general knowledge in the field? State how your results extend the findings of previous studies.
  • Actual version of the results remains presentation of some datasets.
  • In the conclusion is mentioned: “The cutting erosion caused by the small angle….” Please explain / clarify what do you mean by the cutting erosion. Actually I think more appropriate term is material disintegration as is used by eg. Carach in https://doi.org/10.1007/s00170-018-1653-2, with appropriate explanation.
  • Conclusion should be improved​ in term of future direction of research, bonded with discussion part.

Thank you and good luck.

Author Response

Dear reviewer
We have modified or explained the content according to your suggestions. Please refer to the attachment for the specific response.
Thank you.

Reviewer 2 Report

The authors in the current work piece together an analytic model for
studying the erosion damage of a pipe wall by liquid-solid suspension
flow. This model is constructed as a two-phase flow model combined
with a recently developed model for single-particle impact
erosion. Results from the model were then presented under different
conditions and trends discussed under low and high flow rates and
varying pipe diameter and pipe orientation. I particuluarly like when
relatively simple models are capable of capturing some trends as that
opens the possibility for others (including students) to explore
properties of complex systems. While some conclusions are arrived at
based on the developed model, very little is discussed as to how the
trends and observations from this model relate to others in the
field. The authors themselves point out one other multiparticle
erosion prediction model by Huang [10] describing it as having
"numerous considerations and heavy experimental workload...". I was
expecting discussion on Huang's model to also appear near the end of
the manuscript in a comparison with the current model discussing the
strengths and weaknesses. Or comparing results of the current model
with some experimental or simulation data.

The text in Section 2.3 describing the erosion model is essentially
identical to that in section 2 of a recent publication co-authored by
one of the authors [Materials & Technology, 54, 321-326 (2020)] and
could be described as self-plagiarism. I strongly advise rewriting
that section entirely or simply advising the reader to review this
2020 publication for a summary of the erosion model and the earlier
2019 Materials publication for further details.

Some minor corrections: "Figure 6" on line 353 should be "Figure 4" is
discussed in the text before Figure 5.

Author Response

(The authors gave the same response as above.)

Reviewer 3 Report

Dear Authors

Your paper is very well written, in general I don't have any substantive comments. However conclusion are inadequate. The main problem that worries me is whether the theoretical article is related to the subject of the journal, but I leave it to the editors.  

Author Response

Thank you for reviewing our manuscript. 

Erosion, as a form of wear, has always existed in industry.

In this article, we are trying to establish a new method for quickly predicting erosion damage, which is also to reduce wear damage.

Round 2

Reviewer 1 Report

The article entitled "Theoretical prediction method for erosion damage of Horizontal Pipe by Suspended Particles in Liquid-Solid Flow" was significantly improved. Suggestions raised by the reviewers were carefully incorporated. Revised version of the manuscript is more consistent and  can be now published in the journal. 

Reviewer 2 Report

The authors have addressed all of the critiques raised by the
reviewers. In particular, the comparison to experimental data is a
strong addition to the manuscript.